# ROME: Evaluating Pre-trained Vision-Language Models on Reasoning beyond Visual Common Sense

**Kankan Zhou, Eason Lai, Wei Bin Au Yeong, Kyriakos Mouratidis, Jing Jiang**

School of Computing and Information Systems, Singapore Management University

{kkzhou.2020, yblai, wb.auyeong.2021, kyriakos, jingjiang}@smu.edu.sg

## Abstract

Humans possess a strong capability for reasoning beyond common sense. For example, given an unconventional image of a goldfish laying on the table next to an empty fishbowl, a human would effortlessly determine that the fish is not inside the fishbowl. The case, however, may be different for a vision-language model, whose reasoning could gravitate towards the common scenario that the fish is inside the bowl, despite the visual input. In this paper, we introduce a novel probing dataset named **ROME** (**r**easoning bey**o**nd co**m**monsense knowledg**e**) to evaluate whether the state-of-the-art pre-trained vision-language models have the reasoning capability to correctly interpret counter-intuitive content. ROME contains images that defy commonsense knowledge with regards to color, shape, material, size and positional relation. Experiments on the state-of-the-art pre-trained vision-language models reveal that most of these models are still largely incapable of interpreting counter-intuitive scenarios. We hope that ROME will spur further investigations on reasoning beyond commonsense knowledge in vision-language research.

## 1 Introduction

Humans possess not only commonsense knowledge such as *fish are usually found inside (rather than outside) a fishbowl*, but also the ability to handle counter-intuitive scenarios and reason beyond common sense. For example, when presented with the image on the left in Figure 1 and asked "Is the fish inside the fishbowl?" humans would have no problem answering "no." However, it is not clear whether recent pre-trained vision-language models (VLMs) such as LLaVA (Liu et al., 2023) and InstructBLIP (Dai et al., 2023a), which exhibit impressive vision-language understanding abilities, have the capacity to reason beyond visual common sense.

In this paper, we aim to empirically answer this research question by evaluating recent VLMs

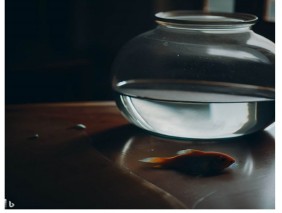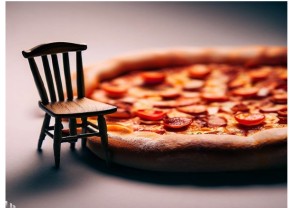

Figure 1: Sample counter-intuitive images from our ROME dataset. Left: an image showing an uncommon *positional relation* between a fish and a fishbowl. Right: an image showing a chair with an uncommon *relative size* with respect to a pizza.

on a new probing dataset we introduce, called ROME (**r**easoning bey**o**nd co**m**monsense knowledg**e**). Such empirical evaluation is important because humans' ability to "think outside the box" is a critical source of creativity and, thus, also a desirable ability for AI models. Unfortunately, most existing benchmarks to evaluate VLMs do not challenge them with unconventional or counter-intuitive situations.

Our ROME dataset consists of 1,563 images generated by DALL-E-2 depicting counter-intuitive scenarios. Instead of relying on a small group of people's notion of counter-intuitive scenarios, we largely leverage a recent visual common sense dataset, ViComTe, that was systematically derived from Visual Genome (Zhang et al., 2022). We automatically convert commonsense assertions related to objects' color, shape, material and size in ViComTe into descriptions of counter-intuitive scenarios, and use DALL-E-2 to generate images accordingly. In addition, we manually use Bing Image Creator to generate counter-intuitive images related to objects' positional relations to supplement the data above.

To assess both a VLM's commonsense knowledge and its abilities to reason beyond common sense, we design two groups of probing tasks in the form of binary (yes/no) visual questions: (1)

Commonsense questions, which are related to the common attribute values of various objects, or the typical relative size or positional relationship between two objects. (2) Counter-intuitive questions, which ask about the objects, their attributes, and their relative size or positional relation in counter-intuitive images. We then define four evaluation metrics based on these probing questions.

We use the ROME dataset and the probing questions to evaluate six pre-trained vision-language models: ALBEF (Li et al., 2021), BLIP-2 (Li et al., 2023a), LLaVA (Liu et al., 2023), MiniGPT-4 (Zhu et al., 2023), mPLUG-Owl (Ye et al., 2023), and InstructBLIP (Dai et al., 2023a). Among these models, BLIP-2, LLaVA, MiniGPT-4, mPLUG-Owl, and InstructBLIP are the latest models that are built on top of pre-trained large language models (LLMs), e.g., LLaMA. They can be directly used by end-users through natural language conversations such as to perform zero-shot visual question answering (VQA) , making them readily available for wide adoption in end-user applications. Therefore, we focus on the evaluation of these state-of-the-art models. In contrast, although earlier VLMs such as CLIP (Radford et al., 2021) and ALBEF also demonstrated strong zero-shot capabilities on tasks including object detection (Radford et al., 2021) and VQA (Song et al., 2022), they usually require careful prompt engineering to work properly, and thus we do not focus on these earlier models. Nevertheless, we still include one of them, namely, ALBEF pre-trained with VQAv2[1], in order to have a comprehensive representation of VLMs.

Our experiments demonstrate that, although these state-of-the-art VLMs are generally effective at counter-intuitive object recognition, most of them perform poorly in terms of counter-intuitive attribute and relation recognition. Meanwhile, interestingly, the models perform poorly on our commonsense questions, often giving inconsistent answers to a pair of questions asking about the same commonsense knowledge. This suggests that many of the latest pre-trained VLMs still lack strong commonsense reasoning capabilities and capabilities to reason beyond common sense, which, in turn, indicates that further improvements are needed in both directions.

---

[1] https://github.com/salesforce/LAVIS

## 2 Related Work

**Visual common sense.** With the rapid development of foundation models, increasing attention is paid to their commonsense reasoning capabilities. Recent studies have investigated the existence of commonsense knowledge in pre-trained language models (e.g., Bouraoui et al., 2020; Lin et al., 2020). Others have focused on visual common sense (e.g., Vedantam et al. (2015); Zellers et al. (2021); Liu et al. (2022)). In particular, Zhang et al. (2022) studied commonsense knowledge pertaining to typical visual attributes of different objects. We follow their notion of visual common sense and use their dataset as our main source of commonsense knowledge. Note that this notion of visual common sense is different from the notion of visual commonsense reasoning in the well-known VCR dataset (Zellers et al., 2019).

While most previous work has focused on commonsense reasoning, we focus on evaluating reasoning capabilities beyond common sense. Bitton-Guetta et al. (2023) introduced WHOOPS, a dataset of weird/unconventional images, where the selection of "weird" scenarios is based on a (relatively) small group of 30 designers' impression of common sense and weirdness. In contrast, our selection of counter-intuitive scenarios is based on the visual commonsense dataset by Zhang et al. (2022) that was systematically constructed. WHOOPS covers a wide range of weird scenarios, many of which involve non-visual common sense, such as social norms, cultural knowledge, and celebrities. On the other hand, our dataset is centered on five categories of common sense that are purely visual. These types of common sense are also more primitive (compared to WHOOPS, where the reasons for weirdness are often complex, involving multiple steps of reasoning). Also, WHOOPS focuses on explanation, while our dataset is framed around the models' ability to overcome common sense and correctly recognitze objects, attributes and spatial relations in counter-intuitive scenarios. Another dataset that contains some counter-intuitive images is the Winoground dataset (Thrush et al., 2022), but their focus is compositional reasoning and not all images in Winoground are counter-intuitive.

**Pre-trained vision-language models.** After the success of BERT (Kenton and Toutanova, 2019), the community started developing pre-trained vision-language models, such as ALBEF (Li

et al., 2021), $X^2$-VLM (Zeng et al., 2023b) and CLIP (Radford et al., 2021), which are trained on web-scale image-text pairs and exhibit powerful zero-shot and few-shot transfer capabilities for downstream tasks. Recently, given the impressive power of LLMs (such as ChatGPT) to perform various tasks through natural language conversations, a new line of work leverages LLMs to build VLMs with parameter-efficient fine-tuning (e.g., BLIP-2 (Li et al., 2023a), LLaVA (Liu et al., 2023) and InstructBLIP (Dai et al., 2023a)). These models usually train an alignment module to convert visual inputs into embeddings acceptable by a pre-trained LLM.

**Evaluation of VLMs.** Recently, there have been various attempts to evaluate VLMs on different aspects. Thrush et al. (2022) constructed the Winoground dataset to assess the ability of VLMs to conduct visio-linguistic compositional reasoning. Zhou et al. (2022) studied the existence of social bias in VLMs. Li et al. (2023b) considered the hallucination problem, i.e., the inclusion of objects that are inconsistent with the given images in the generated text descriptions. Zeng et al. (2023a) compared different LLM backbones and model designs of VLMs to explore the influence of diversified prompts on the instruction-following ability of these models. In contrast, we evaluate state-of-the-art VLMs for reasoning beyond common sense.

## 3 Creation of Counter-intuitive Images

In this section, we explain how we create counter-intuitive images.

### 3.1 Counter-intuitive Descriptions

Counter-intuitive scenarios are rare in real life pictures. Therefore, similar to Bitton-Guetta et al. (2023), we opt to *generate* images using recent text-to-image models. To use these models, we need to come up with textual descriptions of counter-intuitive scenarios, such as *a fish outside a fishbowl*, which needs to be derived from *commonsense assertions* such as *a fish inside a fishbowl*. Therefore, our first step is to identify a large set of visual commonsense assertions.

We use ViComTe (Zhang et al., 2022) as our main source of visual commonsense knowledge, because it is systematically derived from Visual Genome (Krishna et al., 2017), which contains ~10K images from MSCOCO and therefore has a diverse coverage. We use commonsense knowledge related to color, shape, material, and size from ViComTe. We supplement it with a dataset by Liu et al. (2022) (which we call ThingsNotWritten) that contains positional commonsense knowledge.

Color, shape, and material are attributes of physical objects. The commonsense knowledge contained in ViComTe on these attributes is represented as frequencies of each possible attribute value, given an object and an attribute. For example, given the object *knob* and the attribute *shape*, ViComTe provides the frequencies of different shapes such as *round*, *square*, and *triangle* that are associated with knobs. We use only the subset of ViComTe where the most frequent attribute value has at least 80% frequency, i.e., cases where it is common sense for the object to have a dominating attribute value. To create counter-intuitive descriptions, given an object and an attribute (color, shape or material), we choose the attribute value that has the *lowest* frequency (including 0). For example, given the object *tire* and the attribute *color*, the value with the lowest frequency in ViComTe is *blue*. Based on this, we form the counter-intuitive description *a tire in the color of blue*.

ViComTe represents size-related commonsense knowledge as a pair of objects and their relative size, e.g., a deer is *larger than* a plate. We simply reverse the relation to form counter-intuitive descriptions, e.g., a deer that is *smaller than* a plate.

Another important type of visual common sense is the positional relation between two objects, which however is not covered in ViComTe. The ThingsNotWritten dataset by Liu et al. (2022) provides commonsense assertions of this type, e.g., *a fish inside a fishbowl*. Because the size of ThingsNotWritten is relatively small, we include additional positional commonsense assertions based on our own judgment. To convert these positional commonsense assertions into counter-intuitive descriptions, we replace their positional preposition with a random choice from a predefined list in ThingsNotWritten.

For all counter-intuitive descriptions, we experiment with different sentence templates and choose those that work the best with DALL-E-2. More details can be found in Appendix A.

### 3.2 Image Generation

Our preliminary experiments show that for color, shape, and material, automatically generating the

images using DALL-E-2 followed by two rounds of filtering works well (details in §3.3), but for size and positional relation, automatic generation produces too many low-quality images that do not match the given object or attribute descriptions. Hence, for size and positional relation we adopt a manual method.

For automatic generation of counter-intuitive images on color, shape, and material, we feed the counter-intuitive descriptions to DALL-E-2 and generate 20 images per description. We then employ CLIP (Radford et al., 2021) for a first round of filtering to remove low-quality images. Specifically, for each generated image, we use CLIP to calculate the image's cosine similarity with the counter-intuitive description used to generate it, and its cosine similarity with the corresponding commonsense assertion. We then normalize the two similarity scores into probabilities. A threshold of 0.8 is used to discard images whose probability of being counter-intuitive is low. This step reduces the number of images from more than 21K to around 3.5K.

For manual creation of counter-intuitive images on size and positional relation, two researchers experiment with different description templates using Bing Image Creator (which is powered by DALL-E-2) until they achieve the desired high-quality counter-intuitive images.

### 3.3 Human Annotation

For automatically generated images, even after the filtering by CLIP, we find that there are still images where the objects are not recognizable by humans. We therefore conduct a second round of filtering based on human annotations. We split this second round into two steps: object recognition, and attribute confirmation.

In the object recognition step, given an image, we ask three human annotators to identify the depicted object by picking one of three options. The purpose is to make sure that humans can recognize the object, despite its uncommon or weird attributes. For those images where the human annotator is able to recognize the objects, we proceed to the attribute confirmation step, where three human annotators judge whether the object indeed matches the counter-intuitive description used to generate the image. Only images where all three annotators agree on the objects and the attributes are kept. After human filtering, the number of images

| | Color | Shape | Material | Size | Positional |
|---|---|---|---|---|---|
| **Count** | 562 | 310 | 391 | 200 | 100 |

Table 1: Dataset statistics

is further reduced from about 3.5K to 1,263.

To prevent any bias from the human annotators, during the recognition step of filtering, we intentionally introduce data samples where the images do not match the given object descriptions, as a sanity check of the annotators. We calculate Fleiss' kappa for both the object recognition and the attribute confirmation steps. The Fleiss' kappa for the object recognition step is 0.67, indicating substantial agreement. For the attribute confirmation step, the Fleiss' kappa is 0.41, suggesting a moderate level of agreement.

### 3.4 Statistics of ROME

In total, we obtain 1,563 counter-intuitive images. Table 1 provides the breakdown of our dataset into different categories. Some examples of our generated images can be found in Appendix B. ROME will be made publicly available at https://github.com/K-Square-00/ROME.

## 4 Probing Questions and Evaluation Metrics

Our main research question is whether state-of-the-art pre-trained VLMs can handle counter-intuitive images well, i.e., whether they are able to reason beyond common sense. However, we should first address a more fundamental question, which is whether these pre-trained VLMs possess visual common sense in the first place. We therefore create two groups of probing questions: (1) questions testing pre-trained VLMs' visual common sense, and (2) questions testing pre-trained VLMs' reasoning abilities beyond visual common sense.

Our preliminary investigations showed that the latest pre-trained VLMs, such as LLaVA, tend to generate verbose answers to open-ended questions. (See Appendix C for examples.) This makes it hard to automatically process the generated answers. We hence design only binary questions, whose answers can be automatically processed by checking whether they start with "yes" or "no."

### 4.1 Probing Questions

**Commonsense questions.** This set of questions is to test whether a VLM is equipped with common-

| Image | Template | Sample Questions |
|---|---|---|
| **Commonsense question(s)** | **Q1a:** In general, is the color of a/an **[object]** normally **[common attribute]** ? | In general, is the color of a **clementine** normally **orange**? |
| | **Q1b:** In general, is the color of a/an **[object]** normally **[uncommon attribute]** ? | In general, is the color of a **clementine** normally **black**? |
| | **Q2a:** In general, is the color of a/an **[object]** more likely to be **[common attribute]** than to be **[uncommon attribute]** ? | In general, is the color of a **clementine** more likely to be **orange** than to be **black**? |
| | **Q2b:** In general, is the color of a/an **[object]** more likely to be **[uncommon attribute]** than to be **[common attribute]** ? | In general, is the color of a **clementine** more likely to be **black** than to be **orange**? |
| **Counter-intuitive question(s)** — **Object recognition question(s)** | **Q3.:** Does this image contain a/an **[object]** ? | Does this image contain a **clementine**? |
| **Attribute recognition question(s)** | **Q4a:** In this image, is the color of the **[object]** **[common attribute]** ?
**Q4b:** In this image, is the color of the **[object]** **[uncommon attribute]** ? | In this image, is the color of the **clementine** **orange**?
In this image, is the color of the **clementine** **black**? |

Figure 2: Probing question templates and examples of instantiated questions

sense knowledge. We design two pairs of yes/no question templates, as shown in the upper section of Figure 2. The first pair of questions each asks a VLM to judge the truthfulness of an assertion that is either commonsense or counter-intuitive. If the model answers "yes" to Q1a and "no" to Q1b, we deem it to have the corresponding commonsense knowledge. The reason we introduce Q1b is to ensure that models do not score high just by answering "yes" to all questions. The second pair of questions each asks a VLM to compare the likelihood of the commonsense attribute value versus an uncommon attribute value. If the model answers "yes" to Q2a and "no" to Q2b, we deem it to have the respective commonsense knowledge.

**Counter-intuitive questions.** This set of questions are based on the generated counter-intuitive images. Given an image showing a counter-intuitive scenario, we test whether a VLM can overcome visual common sense and answer the questions correctly. Specifically, we design two subgroups of questions: (1) object recognition, and (2) attribute/relation recognition. These question templates are shown in the lower section of Figure 2. Q3 asks about the existence of an object. If a VLM answers "yes" to Q3, the VLM can recognize the object despite its unusual color, shape, or material, or its co-occurrence with another object of unusual relative size or position. This object recognition ability can be regarded as an ability

to reason beyond visual common sense. Q4a and Q4b ask about either the color, shape, or material of the object in the image, or the relative size or positional relation between two depicted objects. Similar to previous probing questions, if the VLM answers "no" to Q4a and "yes" to Q4b, it can correctly identify the counter-intuitive attribute value of the object (or the counter-intuitive relation between two depicted objects), thus demonstrating reasoning abilities beyond visual common sense.

The question templates used are slightly different for different commonsense types. E.g., for *material*, the template Q1b is "In general, is a/an [object] normally made of [uncommon attribute]?" The full list of question templates can be found in Appendix D.

### 4.2 Evaluation Metrics

To facilitate the formal definition of the metrics below, we use $\mathcal{R}$ to denote the set of images in the ROME dataset. For an image $I \in \mathcal{R}$ and a question template $Q$, we use $Q(I)$ to denote the question instantiated from template $Q$ with the objects, attributes and/or relations found in image $I$ as well as the underlying visual commonsense knowledge associated with this image. For example, given an image $I$ showing a blue banana and the knowledge that the commonsense color of bananas is yellow, Q1a($I$) would be "In general, is the color of a banana normally yellow?" and Q1b($I$) would be "In general, is the color of a banana normally blue?"

Let $M$ denote a VLM. We use $M(I_1, Q(I_2))$ to denote the answer given by $M$ for image $I_1$ and question $Q(I_2)$, which is instantiated from template $Q$ with information from image $I_2$. We expect $M(I_1, Q(I_2))$ to be either "yes" or "no".

### 4.2.1 Metrics for Commonsense Knowledge

We introduce two metrics for testing VLMs' commonsense knowledge.

**Commonsense score based on language (CS-L):** When asking a VLM the commonsense probing questions instantiated from templates Q1a, Q1b, Q2a, and Q2b, the model does not have to refer to any image, because the questions are asking about general cases rather than a specific case depicted by any image. Therefore, it is natural to give the VLM a non-informative image, such as a blank image in white or arguably a random image. Let $I_{\text{null}}$ denote this non-informative image. Given model $M$, we define the first metric as follows:

$$\text{CS-L}_1(M) \;=\; \frac{1}{|\mathcal{R}|} \sum_{I \in \mathcal{R}} \text{CS-L}_1(M, I), \quad (1)$$

where $\text{CS-L}_1(M, I)$ is 1 if $M(I_{\text{null}}, \text{Q1a}(I))$ is "yes" and $M(I_{\text{null}}, \text{Q1b}(I))$ is "no". In other words, $\text{CS-L}_1(M, I)$ is 1 if and only if model $M$ answers both $\text{Q1a}(I)$ and $\text{Q1b}(I)$ correctly according to commonsense knowledge, given a non-informative image. $\text{CS-L}_1(M)$ is the average of $\text{CS-L}_1(M, I)$ over all $I \in \mathcal{R}$, i.e., the frequency in which the model demonstrates commonsense knowledge. We calculate $\text{CS-L}_2(M)$ similarly, based on the model's answers to $\text{Q2a}(I)$ and $\text{Q2b}(I)$. The final $\text{CS-L}(M)$ is set to be $max(\text{CS-L}_1(M), \text{CS-L}_2(M))$.

**Commonsense score based on vision and language (CS-VL):** To test whether a VLM might be "distracted" by a counter-intuitive image when given a commonsense question, we define CS-VL just like CS-L but, instead of a non-informative image, we use a counter-intuitive image.

### 4.2.2 Metrics for Reasoning beyond Common Sense

We define two metrics for reasoning beyond visual common sense. One corresponds to object recognition and the other to attribute/relation recognition.

**Counter-intuitive score based on object recognition (CI-Obj):** This metric is designed based on the motivation that if a VLM cannot recognize a counter-intuitive object, then it has limited ability to reason beyond common sense. Given an image

$I$ that contains $k$ objects (where $k$ is 1 for images related to color, shape, and material, and $k$ is 2 for images related to size and positional relation), $\text{Q3}(I)$ is a set of $k$ questions, each corresponding to an object in $I$. We define $\text{CI-Obj}(M, I)$ to be 1 if and only if model $M$ can recognize all objects in $I$. In turn, $\text{CI-Obj}(M)$ is the average of $\text{CI-Obj}(M, I)$ over all images.

**Counter-intuitive score based on attribute/relation recognition (CI-AttrRel):** If a VLM cannot recognize counter-intuitive attributes or relations, its reasoning beyond common sense is deemed poor. Given a counter-intuitive image $I$, we define $\text{CI-AttrRel}(M, I)$ to be 1 if and only if model $M$ answers both $\text{Q4a}(I)$ and $\text{Q4b}(I)$ correctly, according to the actual content of image $I$. $\text{CI-AttrRel}(M)$ is the average of $\text{CI-AttrRel}(M, I)$ over all $I \in \mathcal{R}$.

## 5 Experiments

### 5.1 VLMs for Evaluation

We focus on six of the latest VLMs, summarized in Table 2. The first five leverage LLMs with parameter-efficient fine-tuning and demonstrate zero-shot capabilities on downstream tasks. For completeness, we also consider ALBEF fine-tuned on VQAv2. Some of these models tend to give verbose answers, even to binary questions. We hence treat an answer as "yes" (or "no") if it *starts with* "yes" (or "no"). In the penultimate column of Table 2, we show the percentage of such answers. Except for LLaVA, the percentage is high.

### 5.2 Main Results

Table 3 presents the performance of the models over the entire ROME dataset, according to the four metrics in §4.2. We use blank images as non-informative images to obtain CS-L. We can answer the following research questions based on the results.

**(1) When given a non-informative image, do state-of-the-art VLMs demonstrate visual commonsense knowledge?**
The relevant metric for this question is CS-L. We can see that the CS-L scores of all models are relatively low, ranging from 10.89% to 32.31%. This indicates that they do not exhibit strong visual commonsense knowledge. This is to some extent unexpected, because previous work reported that CLIP and Oscar, two earlier pre-trained VLMs, achieved between 60% and 80% of accuracy when predict-

| Model | Visual Encoder | Alignment Network | LLM | Yes/No Rate (%) | Hallucination Rate (%) |
|---|---|---|---|---|---|
| **BLIP-2** | ViT-L/14 | Q-Former | $OPT_{2.7B}$ | 99.81 | 30 |
| **InstructBLIP** | ViT-L/14 | Q-Former | $Vicuna_{7B}$ | 95.91 | **5** |
| **LLaVA** | ViT-L/14 | Linear | $LLaMA_{7B}$ | 39.22 | 100 |
| **MiniGPT-4** | ViT-G/14 | Linear | $Vicuna_{7B}$ | 99.81 | 40 |
| **mPLUG-Owl** | ViT-G/14 | Attention | $LLaMA_{7B}$ | 98.78 | 65 |
| **ALBEF** | ViT-B/16 | Attention | $BERT_{base}$ | 100.00 | **5** |

Table 2: The VLMs included in the evaluation

| Model | Commonsense | | Counter-intuitive | |
| | CS-L | CS-VL | CI-Obj | CI-AttrRel |
|---|---|---|---|---|
| **BLIP-2** | 5.82 | 2.48 | 91.88 | 27.80 |
| **InstructBLIP** | **32.31** | 9.66 | 94.75 | **63.72** |
| **LLaVA** | 31.41 | **28.09** | **98.34** | 0.13 |
| **MiniGPT-4** | 10.89 | 4.41 | 94.56 | 5.31 |
| **mPLUG-Owl** | 28.53 | 18.11 | 97.38 | 35.83 |
| **ALBEF** | 14.01 | 0.51 | 90.79 | 44.53 |

Table 3: Overall evaluation results (scores in %)

ing the commonsense color, shape, and material of objects (Zhang et al., 2022). We therefore further inspect the results and find that the mistakes made by the VLMs we are evaluating are usually due to inconsistent responses. For example, in Figure 3, when asked to compare the relative size of a car and a towel, mPLUG-Owl gives contradictory answers to the two probing questions. Similar inconsistencies are commonly found with all models.

Although blank images are non-informative, one may argue that blank images are out-of-distribution data considering that these pre-trained VLMs have rarely encountered blank images in their training data. Therefore, choosing blank images might affect these models' performance. To address this concern, we also conduct additional experiments with randomly selected non-blank images. Specifically, for each instance in our dataset, we repeat the probing experiments for the CS-L scores, but instead of a blank image, we use a randomly selected image from the Visual Genome dataset. We repeat the process three times on all models. The results are shown in Table 5. The results indicate that using randomly selected images produces similar CS-L scores to using blank images.

Overall, we find that InstructBLIP and LLaVA answer commonsense questions correctly about 30% and 35% of the time, regardless of whether a blank image or a random image is given. For AL-BEF, BLIP-2, MiniGPT-4 and mPLUG-Owl, using randomly selected images produces slightly lower CS-L scores than using blank images (around 5%

difference), implying that these models are prone to visual distractions caused by the random images not related to the questions.

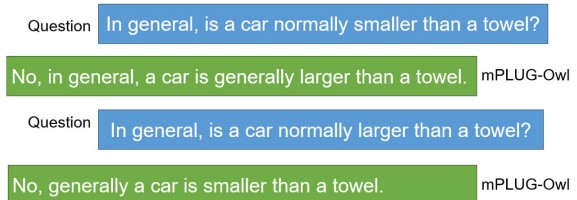

Figure 3: An example of inconsistent answers to two questions asking about the same commonsense knowledge. A blank image is input to mPLUG-Owl.

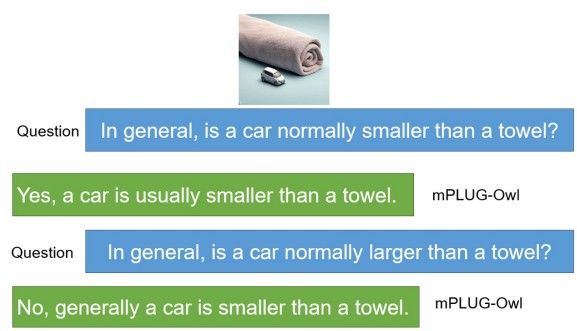

Figure 4: An example of how a counter-intuitive image weakens commonsense reasoning

**(2) When presented with counter-intuitive images and asked about commonsense knowledge, do the models behave differently?**

Comparing CS-L with CS-VL, it is clear that once the counter-intuitive images are presented, the commonsense reasoning of all models is further impaired; the CS-VL scores are lower, ranging between 0.51% and 28.09%. That is not surprising because the counter-intuitive images may confuse VLMs during commonsense reasoning. For example, in Figure 4, when mPLUG-Owl is given a counter-intuitive image, it answers incorrectly even the first commonsense question (although it did get it right in Figure 3 with a blank image input). This problem is widely observed with all models.

| Model | Color | Shape | Material | Size | Position |
|-------|-------|-------|----------|------|----------|
| **BLIP-2** | 46.26 | 2.00 | 32.23 | 3.00 | 8.00 |
| **InstructBLIP** | 84.34 | 28.39 | 57.03 | 88.00 | 35.00 |
| **LLaVA** | 0.18 | 0.00 | 0.00 | 0.00 | 1.00 |
| **MiniGPT-4** | 8.01 | 0.00 | 7.93 | 3.50 | 0.00 |
| **mPLUG-Owl** | 50.89 | 30.00 | 33.76 | 17.50 | 14.00 |
| **ALBEF** | 50.00 | 1.94 | 60.36 | 70.00 | 33.00 |

Table 4: CI-AttrRel (in %) for different commonsense types

| Model | Commonsense CS-L | |
|-------|------------------|---|
| | (Blank Image) | (Random Image) |
| **BLIP-2** | 5.82 | 0.30 |
| **InstructBLIP** | **32.31** | **35.03** |
| **LLaVA** | 31.41 | 31.01 |
| **MiniGPT-4** | 10.89 | 6.07 |
| **mPLUG-Owl** | 28.53 | 25.39 |
| **ALBEF** | 14.01 | 8.79 |

Table 5: Blank image vs random image (scores in %)

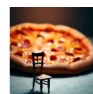

> **In this image, is the chair larger than the pizza?**
>
> **BLIP-2:** Yes
> **InstructBLIP:** No, the chair is smaller than the pizza.
> **LLaVA:** In the image, the chair is not larger than the pizza.
> **MiniGPT-4:** Yes, the chair is larger than the pizza.
> **mPLUG-Owl:** Yes, the chair is bigger than the pizza in the image.

> **In this image, is the chair smaller than the pizza?**
>
> **BLIP-2:** No
> **InstructBLIP:** Yes, the chair is smaller than the pizza.
> **LLaVA:** In this image, the chair is not smaller than the pizza.
> **MiniGPT-4:** Yes, the chair is smaller than the pizza.
> **mPLUG-Owl:** Yes, in the image, the chair is smaller than the pizza.

Figure 5: Examples of models' responses to counter-intuitive questions

**(3) When given counter-intuitive images, can the models correctly recognize the depicted objects, even though these objects are shown with counter-intuitive attributes or have a counter-intuitive relative size or counter-intuitive spatial position with respect to other objects?**
Surprisingly, all models perform very well in counter-intuitive object recognition. They achieve CI-Obj scores from 90.79% to 98.34%, which suggests a positive answer to this research question.

That said, recent work has shown that some VLMs may suffer from the hallucination problem (Li et al., 2023b; Dai et al., 2023b), i.e., they may determine that an object exists in an image while in reality it does not. This may affect the interpretation of our results, because if a VLM tends to answer "yes" when asked about the existence of *any* object, even when that object is not in the

image, our conjecture above may not hold. To get a sense of how common the hallucination problem is, we first randomly retrieve 20 images containing different objects from ROME. Next, for each image we pick a non-exist-object label at random from the exist-object labels of the other 19 images. Once all images are assigned a non-exist-object label, we ask "Does the image contain a/an [non-exist-object]?" The hallucination rate is presented in Table 2. All models suffer from object hallucination to some extent, but LLaVA and mPLUG-Owl suffer the most. On the other hand, the hallucination rate of InstructBLIP and ALBEF is only 5%. The results suggest that for InstructBLIP and ALBEF, we can safely claim that they can recognize counter-intuitive objects well, for BLIP-2 and MiniGPT-4, they probably can recognize counter-intuitive objects well, but for LLaVA and mPLUG-Owl, we cannot draw conclusions.

**(4) Can the models recognize counter-intuitive attributes and spatial relations?**
The relevant metric here is CI-AttrRel. The models' scores are low, except maybe for InstructBLIP, which achieves the highest CI-AttrRel of 63.72%. We infer that, despite recognizing the objects successfully, VLMs struggle with recognizing counter-intuitive attributes and spatial relations. Figure 5 presents all models' responses when the image depicts a pizza that is larger than a chair. BLIP-2 answers both questions wrongly. MiniGPT-4 and mPLUG-Owl answer "yes" to both questions and are thus inconsistent. LLaVA's answers do not start with "yes" or "no" but judging from the complete responses, it contradicts itself. This behaviour suggests that the models have room for improvement in terms of both visual language understanding and logical consistency. On the other hand, in this example InstructBLIP answers both questions correctly.

In Table 4, we break down the models' CI-AttrRel scores for each commonsense type. Although the models perform very differently, recog-

| Model | CI-AttrRel | CM-AttrRel |
|---|---|---|
| **InstructBLIP** | 63.72 | 77.00 |
| **LLaVA** | 0.13 | 4.00 |

Table 6: CI-AttrRel vs CM-AttrRel (in %)

nizing counter-intuitive colors turns out to be the easiest for most of them. For BLIP-2, MiniGPT-4 and mPLUG-Owl, recognizing counter-intuitive material is the second easiest. Interestingly, for InstructBLIP and ALBEF, size is the easiest, whereas for the other models it tends to be very hard.

### 5.3 Further Investigation on Common Sense

The finding that the VLMs do not demonstrate strong visual commonsense knowledge appears contradictory to previous reports with CLIP and Oscar (Zhang et al., 2022). We have pointed out in §5.2 that a main reason we find is that the VLMs we evaluate often give inconsistent answers. We therefore suspect that the low CS-L scores may not be due to VLMs lacking commonsense knowledge per se, but maybe the phrasing of our questions does not "activate" that knowledge. To investigate this, we pick one of the VLMs, LLaVA, and further test it with the following open-ended question: "In general, what is the common [attribute] of a/an [object]?" where [attribute] is set to color, shape or material. We experiment with 20 questions. Overall, LLaVA is able to give meaningful commonsense answers 75% of the time. This observation confirms our suspicion that some VLMs' commonsense reasoning ability may be sensitive to the way the question is phrased. We believe that this may be a general weakness of the latest VLMs.

Our experiments earlier on CI-AttrRel focus on attribute recognition in counter-intuitive images. It would be helpful to check how well the VLMs perform attribute recognition in *commonsense* images. We therefore also conduct another set of additional experiments to test whether attribute recognition is easier for these VLMs in commonsense images. In this set of experiments, we randomly sample 100 images from ROME from the color, shape, and material categories and create 100 corresponding commonsense images (e.g., yellow bananas) via Bing Image Creator. We choose LLaVA and InstructBLIP and perform the same experiments as for CI-AttrRel, albeit with commonsense images as visual input. We report the respective score, denoted as CM-AttrRel, in Table 6. We observe that

when the models are presented with commonsense images and asked about commonsense attributes (e.g., "In this image, is the color of the banana yellow?"), they clearly perform better (i.e., give consistent commonsense answers) than when presented with counter-intuitive images. This is not surprising, because the commonsense images present visual commonsense knowledge, which presumably aligns with the commonsense knowledge implicitly contain in the VLMs.

## 6 Conclusion

In this work, we construct the ROME dataset and propose a probing framework for evaluating the capability of pre-trained vision-language models (VLMs) to reason beyond common sense. Using ROME and a set of metrics we define, we show that state-of-the-art VLMs are lacking in this aspect. We hope ROME will spur further research in the important direction of reasoning beyond common sense in the NLP and vision communities.

## 7 Limitations

Generation of counter-intuitive images requires significant human involvement, as current state-of-the-art text-to-image generation models, such as DALL-E-2, have a strong bias towards commonsense images. Such human involvement may inevitably introduce human bias to our dataset, including possible cultural bias because all our annotators come from Asia. Additionally, common sense may not be binary. There are degrees to which one might consider something to be of common sense, hence, selecting an unequivocally non-commonsense characteristic for an object is also a challenge.

We designed only binary questions to probe VLMs, which may not "activate" the full knowledge and capabilities of VLMs. For example, as discussed earlier, some VLMs may show commonsense knowledge when prompted in a different way. We leave the investigation of how to more effectively prompt VLMs to understand their commonsense-related capabilities, and why VLMs behave in inconsistent ways when they are given the same questions phrased differently, as our future work.

## Acknowledgements

This research was supported by the Ministry of Education, Singapore, under its Academic Re-

search Fund Tier 2 (Proposal ID: T2EP20222-0047, Project ID: MOE-000440-00). Eason Lai and Kyriakos Mouratidis were supported by the Ministry of Education, Singapore, under its Academic Research Fund Tier 2 (Award No. MOE-T2EP20121-0002). Any opinions, findings and conclusions or recommendations expressed in this material are those of the authors and do not reflect the views of the Ministry of Education, Singapore.

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

## A    Examples of Templates

In Figure 6, we lists some examples of templates that we use to instruct DALL-E-2 to generate images.

| Color |
| --- |
| A full photo of a/an {attribute value} {object}. |
| A photo of {object} in color of {attribute value}. |

| Material |
| --- |
| A/an {object} made of/with {attribute value}. |
| A photo of a/an {attribute value} {object}. |
| A photo of {object} in material of {attribute value}. |
| A/an {object} made of/with {attribute value}, digital art. |
| A photo of {object} in material of {attribute value}, digital art. |

| Shape |
| --- |
| A full photo of {attribute value} {object}. |
| A photo of a/an {object} in the shape of {attribute value}. |
| A full photo of a/an {attribute value} {object}, digital art. |
| A photo of {attribute value} {object}. |
| A photo of {object} in the shape of {attribute value}, digital art. |

Figure 6: Examples of templates

## B    Sample Images in ROME

In Figure 7, we show examples of the generated images that have been validated by human annotators and included in the final ROME dataset.

## C    Open-ended Questions

In Figure 8, we show the responses from LLaVA to some open-ended questions. These responses suggest that if we were to raise open-ended questions instead of binary (i.e., yes/no) questions, it would be very difficult to automatically extract the answer to these questions. Therefore, in our design of probing questions, we opt for binary questions.

## D    Probing Questions

Figure 9 shows the full list of probing questions we have designed based on the different types of commonsense knowledge we intend to probe. For some pre-trained VLMs (specifically, InstructBLIP, LLaVA, and mPLUG-Owl), we find that adding the following extra instruction helps: "Please answer yes or no." Therefore, in our experiments, we include this extra instruction for these models.

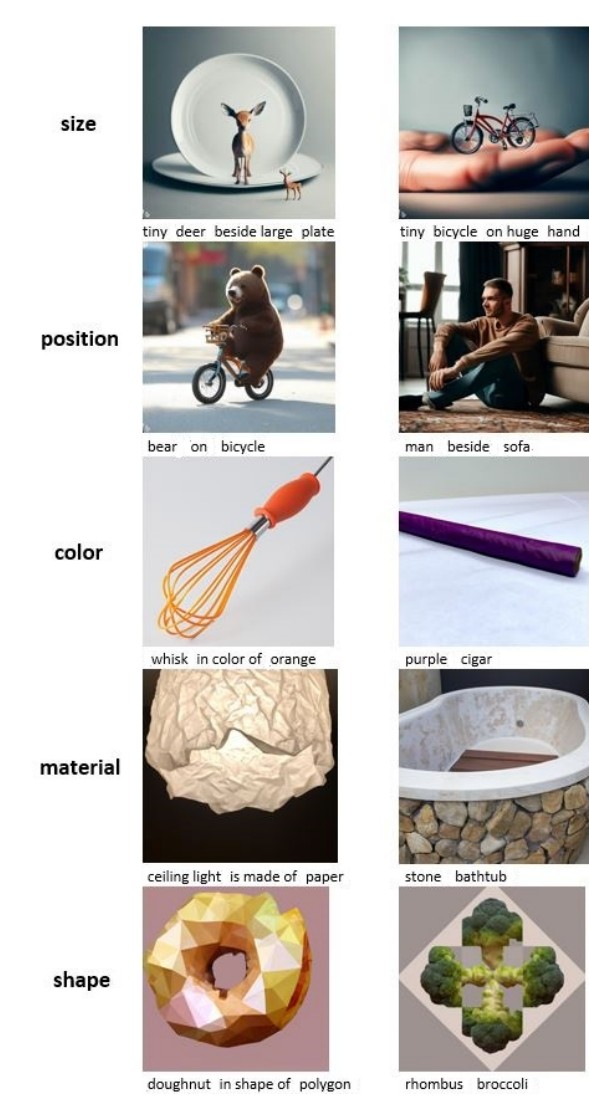

Figure 7: ROME Sample Images

| Open-ended commonsense question | LLaVA response |
|---|---|
| In general, what is the shape of a/an door knob? | The shape of a door knob is typically round or oval, with a flat or slightly curved surface and a hole in the center to fit a doorknob handle. |
| In general, what is the color of a/an chair arm? | The color of a/an chair arm depends on its design and materials. Chair arms can be made from various materials such as wood, metal, or plastic, and they can come in a variety of colors. |
| In general, what is the color of a/an knife handle? | The color of a knife handle can vary depending on the specific type of knife and its design. However, in general, knife handles can be found in a range of colors, including black, silver, gold, and other metallic shades. Some knives may also have wooden or plastic handles. |
| In general, what is the color of a/an burger? | A burger typically has a combination of ingredients that include a cooked patty of ground beef, a bun, and various toppings such as lettuce, tomato, cheese, and condiments like ketchup, mustard, or mayonnaise. |
| In general, what is the water drain made of? | The water drain is typically made of materials like stainless steel, cast iron, or plastic, which are designed to allow water to flow through while preventing clogging. |
| In general, what is the hand soap made of? | Hand soap is typically made from a combination of ingredients such as water, soap base (usually a sodium or potassium soap), and various optional ingredients like fragrances, moisturizers, and coloring agents. The soap base is mixed with water and other ingredients, and then it undergoes a chemical reaction to create the soap bubbles that clean hands and provide a refreshing feel. The fragrances and moisturizers added to the soap can enhance the user's experience, making the handwashing process more enjoyable and providing additional benefits like moisturizing and deodorizing the skin. |

Figure 8: Sample responses from LLaVA for open-ended questions.

| | Type of commonsense knowledge | | | | |
|---|---|---|---|---|---|
| | color | shape | material | size | position |
| **Commonsense Question(s)** | In general, is the color of a/an [object] normally [common attribute] ? | In general, is the shape of a/an [object] normally [common attribute] ? | In general, is a/an [object] normally made of [common attribute] ? | In general, is a/an [object 1] normally larger than a/an [object 2] ? | In general, is a/an [object 1] normally [common relation] a/an [object 2]? |
| | In general, is the color of a/an [object] normally [uncommon attribute] ? | In general, is the shape of a/an [object] normally [uncommon attribute] ? | In general, is a/an [object] normally made of [uncommon attribute] ? | | In general, is a/an [object 1] normally [uncommon relation] a/an [object 2]? |
| | In general, is the color of a/an [object] more likely to be [common attribute] than to be [uncommon attribute] ? | In general, is the shape of a/an [object] more likely to be [common attribute] than to be [uncommon attribute] ? | In general, is a/an [object] more likely to be made of [common attribute] than to be made of [uncommon attribute] ? | In general, is a/an [object 1] normally smaller than a/an [object 2] ? | In general, is a/an [object_1] more likely to be [common relation] a/an [object 2] than to be [uncommon relation] a/an [object 2] ? |
| | In general, is the color of a/an [object] more likely to be [uncommon attribute] than to be [common attribute] ? | In general, is the shape of a/an [object] more likely to be [uncommon attribute] than to be [common attribute] ? | In general, is a/an [object] more likely to be made of [uncommon attribute] than to be made of [common attribute] ? | | In general, is a/an [object_1] more likely to be [uncommon relation] a/an [object 2] than to be [common relation] a/an [object 2] ? |
| **Object Recognition Question(s)** | Does this image contain a/an [object] ? | Does this image contain a/an [object] ? | Does this image contain a/an [object] ? | Does this image contain a/an [object 1] ? | Does this image contain a/an [object 1] ? |
| | | | | Does this image contain a/an [object 2] ? | Does this image contain a/an [object 2] ? |
| **Counter-Intuitive Question(s)** | In this image, is the color of the [object] [common attribute] ? | In this image, is the shape of the [object] [common attribute] ? | In this image, is the [object] made of [common attribute] ? | In this image, is the [object 1] larger than the [object 2] ? | In this image, is the [object 1] [common relation] the [object 2]? |
| | In this image, is the color of the [object] [uncommon attribute] ? | In this image, is the shape of the [object] [uncommon attribute] ? | In this image, is the [object] made of [uncommon attribute] ? | In this image, is the [object 1] smaller than the [object 2] ? | In this image, is the [object 1] [uncommon relation] the [object 2]? |

Figure 9: The full list of probing questions by the types of commonsense knowledge.