# OpenReview forum: "ROME: Evaluating Pre-trained Vision-Language Models on Reasoning beyond Visual Common Sense"
_EMNLP/2023/Conference — EMNLP 2023 Findings_

### Official Review · Reviewer_8GUZ · 2023-08-04

**Soundness:** 3

**Excitement:**

4: Strong: This paper deepens the understanding of some phenomenon or lowers the barriers to an existing research direction.

**Paper Topic And Main Contributions:**

This paper collects a sizable dataset that focuses on counter-intuitive commonsense reasoning. Specifically, they perform an algorithmic (heuristics-based) approach to generate counter-intuitive descriptions (mostly among objects) and leverage generative models such as DALLE-2 (with manual and algorithmic refinements) to generate imageries based on these descriptions. They evaluate numerous state-of-the-art vision-language models on their benchmarks, suggesting significant rooms of improvement, especially in the counter-intuitive attribute and relation recognition of objects.

**Questions For The Authors:**

- What was the scheme used to decide whether to retain a generated image or not?
- Have you considered using some simulated engines (such as Unity) and simply programmatically render the imageries, which can be even more scalable for certain reasoning types (e.g., sizes).
- For non counter-intuitive commonsense probing, VCR [1] and relevant tasks should’ve already investigated quite a decent amount of effort. How does the probing in this work (on this matter) bring more insights?

[1] Zellers, Rowan, et al. "From recognition to cognition: Visual commonsense reasoning." CVPR-19.

**Reasons To Accept:**

- The proposed ROME dataset would be a useful testbed for evaluating the models’ capabilities of grasping the counter-intuitive reasoning on common objects.
- The implementation of leveraging both ViComTe and image generative models is neat and should be well-extendable.
- The conducted experiments are comprehensive and solid, on various models as well as different aspects of their systematic probing using the proposed testbed.

**Reasons To Reject:**

- The inter-annotator agreements on the human annotations need to be reported.
- Following the previous point, the method of retaining the samples needs to be clearly written, i.e., it is unclear to me whether a majority vote is conducted, or a consensus vote is.
- From Section 5.3, if there are indeed better prompts or ways of asking the same questions with different query formats to the VLMs, why did the authors not just follow similar schemes to design both the commonsense and the counter-intuitive questions?

**Reproducibility:**

3: Could reproduce the results with some difficulty. The settings of parameters are underspecified or subjectively determined; the training/evaluation data are not widely available.

**Reviewer Confidence:**

4: Quite sure. I tried to check the important points carefully. It's unlikely, though conceivable, that I missed something that should affect my ratings.

**Typos Grammar Style And Presentation Improvements:**

- L440-L450 is not really necessary. Simply describing that the metric requires the model to be correct in both directions is sufficient and well-justified.
- Same as above, Section 4.2.2 could’ve simply just described the terms of the metrics, without those over-specified notations.

---

> ### Author Rebuttal · Authors · 2023-08-29
>
> ### Inter-annotator agreements
>
> We thank the reviewer for this suggestion and will report these when revising the paper.
>
>
> ### The method / scheme of retaining the samples
>
> We will briefly explain the scheme of retaining the samples here and will include a clear description of the procedure when revising the paper.
>
> We have two kinds of images: (1) those automatically generated by DALL-E-2 API, and (2) those manually and iteratively created by Bing Image Creator. For images generated by DALL-E-2, we go through three sequential rounds of filtering: (a) an automatic filtering step using CLIP, (b) a manual filtering step to check whether the object in the image can be recognized by humans, and (c) another manual filtering step to check whether the counter-intuitive attribute in the image can be recognized by humans. An image is retained only after it passes all three rounds of filtering.
>
> ### Why commonsense and counter-intuitive questions are different
>
> The commonsense questions and counter-intuitive questions shown in Figure 2 in the paper are different because commonsense questions are testing the models’ understanding of the common attributes or spatial relations of some objects, rather than the attributes or relations shown in a specific image. Therefore, we use “in general” to start these questions and we do not mention “this image” in the question. Counter-intuitive questions, on the other hand, have to refer to the specific image given. Therefore, we use “in this image” in the question.
>
> ### Simulated engines such as Unity
>
> We thank the reviewer for introducing Unity. We were not aware of these simulated engines and we will explore the possibility of using them in our future work.
>
> ### New insights compared with VCR and relevant tasks
>
> For non-counter-intuitive probing, we agree that there has been much previous work. The main difference between our work and previous work in terms of non-counter-intuitive probing is that we perform analysis and offer insights on the latest VLMs, which are built on top of LLMs. Given that these models are very recent, to the best of our knowledge, there has not been any systematic study of their visual commonsense capabilities.

---

### Official Review · Reviewer_2QPL · 2023-08-05

**Typos Grammar Style And Presentation Improvements:** n/a
**Soundness:** 3

**Excitement:**

3: Ambivalent: It has merits (e.g., it reports state-of-the-art results, the idea is nice), but there are key weaknesses (e.g., it describes incremental work), and it can significantly benefit from another round of revision. However, I won't object to accepting it if my co-reviewers champion it.

**Missing References:**

[1] What Matters in Training a GPT4-Style Language Model with Multimodal Inputs?
https://arxiv.org/abs/2307.02469

[2]X2-VLM: All-In-One Pre-trained Model For Vision-Language Tasks
https://arxiv.org/abs/2211.12402

**Paper Topic And Main Contributions:**

This paper introduce a new dataset named ROME to evaluate the reasoning capability of VLMs to correctly interpret counter-intuitive content. This dataset contains images that defy commonsense knowledge with regards to color, shape, material, size and positional relation. Experiments on the state-of-the-art multimodal LLMs reveal that most of these models are still largely incapable of interpreting counter-intuitive scenarios.

**Questions For The Authors:**

n/a

**Reasons To Accept:**

The proposed dataset is new and challenging, and can effectively evaluate VLMs' capability in image understanding.

**Reasons To Reject:**

1) The assessment is exclusively conducted on multimodal LLMs, which may not be sufficient. Recent study[1] indicates that most of these models have limited capabilities in image comprehension. For example, these models generate inaccurate image descriptions. In contrast, general VLMs such as ALBEF as mentioned and X-VLM[2] if fine-tuned on VQA2.0 very likely perform better than these multi-modal LLMs on the proposed task.


2) This paper introduces a dataset, however, without proposing any new techniques or tricks to enhance VLMs' efficacy in this task.

3) The proposed dataset is tiny (only 1k images). Similar samples are likely to be included in larger dataset such as VQA2.0 or GQA, with evaluation on these large datasets yielding more comprehensive results.

**Reproducibility:**

3: Could reproduce the results with some difficulty. The settings of parameters are underspecified or subjectively determined; the training/evaluation data are not widely available.

**Reviewer Confidence:**

4: Quite sure. I tried to check the important points carefully. It's unlikely, though conceivable, that I missed something that should affect my ratings.

---

> ### Author Rebuttal · Authors · 2023-08-29
>
> ### Lack of evaluation on other VLMs
>
> We thank the reviewer for this valuable comment and we acknowledge that this is a good suggestion. We have conducted additional experiments with ALBEF fine-tuned on VQAv2 [1], and our experiments show that fine-tuned ALBEF has limited commonsense knowledge but is comparable to multimodal LLMs for counter-intuitive reasoning. Details are given below. We will include evaluation on other VLMs such as X-VLM when revising the paper.
>
> For the experiments with fine-tuned ALBEF, the results can be found in the table below in the last row. The other rows are taken from the original submission and are shown for comparison purposes. We can make the following observations: (1) When fine-tuned ALBEF is presented with counter-intuitive images, it shows strong object recognition abilities (90.63%), at a level close to BLIP-2 and InstructBLIP. (2) When fine-tuned ALBEF is presented with counter-intuitive images, its abilities to recognize attributes and spatial relations (which are counter-intuitive) (47.01%) are weaker than InstructBLIP (67.26%) but stronger than the other multimodal LLMs we evaluated. As the reviewer suspected, ALBEF indeed has some advantage over many multimodal LLMs in this aspect, although it still loses out to InstructBLIP. (3) When fine-tuned ALBEF is presented with either a blank image or a counter-intuitive image and prompted with commonsense questions, its performance (12.67% on blank images and 0.36% on counter-intuitive images) is substantially lower than some of the multimodal LLMs such as InstructBLIP, LLaVA, and mPLUG-Owl. We suspect that this is because ALBEF does not leverage a large pre-trained LLM, whereas InstructBLIP, LLaVA and mPLUG-Owl make use of pre-trained LLMs such as LLaMA, and their commonsense knowledge is likely derived mostly from the LLMs.
>
> |              | Counter-intuitive | Counter-intuitive | Commonsense | Commonsense |
> |:------------:|:-----------------:|:-----------------:|:-----------:|:-----------:|
> |   **Model**  |     **CI-Obj**    |   **CI-AttrRel**  |   **CS-L**  |  **CS-VL**  |
> |    BLIP-2    |       88.76%      |       25.69%      |    14.45%   |    2.50%    |
> | InstructBLIP |       92.77%      |     **67.26%**    |  **39.96%** |    23.55%   |
> |     LLaVA    |     **98.22%**    |       0.18%       |    30.06%   |  **27.92%** |
> |   MiniGPT-4  |       93.93%      |       5.08%       |    14.63%   |    5.62%    |
> |   mPLUG-Owl  |       97.23%      |       35.33%      |    28.37%   |    17.40%   |
> |     ALBEF    |       90.63%      |       47.01%      |    12.67%   |    0.36%    |
>
> ### No new techniques proposed to enhance VLMs
>
> We would like to emphasize that the main contributions of our work are the benchmark probing dataset and the analysis of existing models for reasoning capabilities beyond visual common sense, which to the best of our knowledge have not been studied, especially on the latest multimodal LLMs. There is a line of similar work focusing on constructing benchmark datasets without trying to improve the models, such as [2] and [3].
>
> To answer the reviewer’s question on possible techniques to enhance VLMs, an option could be to artificially introduce counter-intuitive images into the training data of these models, but this approach may be too demanding in terms of computing resources. Another approach would be to perform parameter-efficient fine-tuning of these models using counter-intuitive images. This would be our future work to explore.
>
> ### Size of the dataset
> Please note that VQA2.0 and GQA are commonly used for QA model training, whereas the purpose of our dataset is to evaluate pre-trained VLMs’ zero-shot performance on the tasks we defined. For our purposes (i.e., benchmarking, not training), it is not important for the probing dataset to be at the scale of VQA2.0 and GQA. Please also note that recent similar probing datasets have similar sizes to ours. For example, the Winoground dataset [2] has 1600 images. The WinoGAViL dataset [3] has 3500 images. Probing datasets cannot easily scale up because the data typically require human annotation.
>
> Also, regarding the comment that “similar samples are likely to be included in larger dataset such as VQA2.0 or GQA”, we would like to emphasize that the images in VQA2.0 and GQA are mostly common images whereas our dataset contains counter-intuitive images, which are not easy to be found in VQA2.0 and GQA.
>
> ### Missing references
>
> Thank you for pointing out these missing references. We will include them when revising the paper.
>
>
>
>
> [1] https://github.com/salesforce/LAVIS
>
> [2] Thrush et al., “Winoground: Probing Vision and Language Models for Visio-Linguistic Compositionality,” in CVPR 2022.
>
> [3] Bitton et al., “WinoGAViL: Gamified Association Benchmark to Challenge Vision-and-Language Models,” in NeurIPS 2022.

---

### Official Review · Reviewer_EaB5 · 2023-08-12

**Typos Grammar Style And Presentation Improvements:** Line 563
**Soundness:** 4

**Excitement:**

4: Strong: This paper deepens the understanding of some phenomenon or lowers the barriers to an existing research direction.

**Paper Topic And Main Contributions:**

The paper introduces a partially machine-generated dataset called ROME with over 1K samples to evaluate Vision-Language Models (VLMs)'s capability to interpret counter-intuitive content related to color, shape, material, size and positional relation.

**Questions For The Authors:**

Question A: Can you justify the use of a blank image from a ML perspective as the model and its vision encoder hasn't been trained on this distribution. Does it contribute to the low score?
Question B: Can you provide some preliminary results on the models performance when presented with a common image in attributes and spatial relations? And how does that contrast with the performance with counter-intuitive images?
Question C: Will you open-source the dataset and its pipeline? I am currently marking a 3 on reproducibility and is happy to adjust it to 5 if so.

**Reasons To Accept:**

1. The task to evaluate VLMs under counter-intuitive content is interesting and the dataset can be a helpful resource to the community to access VLM's capabilities.
2. The conclusions this paper derives are valuable and raises many interesting questions. Eg: how to improve robustness of VLMs under prompt variation, how to improve their capabilities under counter-intuitive content.

**Reasons To Reject:**

1. The experiment's methodology of feeding a blank image into the model is flawed, as this blank-image input is entirely out-of-distribution compared to the training data for all five models. This makes the argument that "This indicates that they do not exhibit strong visual commonsense knowledge" less convincing.
2. The paper asserts limited performance under counter-intuitive scenarios with respect to attributes and spatial relations without providing a comparative analysis of the model's performance in typical environments. This lack of comparison makes it difficult to assess the true impact of counter-intuitive content on the model's reasoning capabilities.

**Reproducibility:**

5: Could easily reproduce the results.

**Reviewer Confidence:**

3: Pretty sure, but there's a chance I missed something. Although I have a good feel for this area in general, I did not carefully check the paper's details, e.g., the math, experimental design, or novelty.

---

> ### Author Rebuttal · Authors · 2023-08-29
>
> ### Question A
>
> We thank the reviewer for the question on the use of blank images. We agree that blank images are considered out-of-distribution data. Our rationale for using blank images is the following: (1) Blank images do not contain any visual information and presumably would not mislead the vision-language model in any way, thus allowing us to assess the model’s commonsense knowledge. (2) Ideally, pre-trained large vision-language models are expected to be robust and able to handle out-of-distribution data, including blank images. In fact, the counter-intuitive images we created are also out-of-distribution images.
>
> That said, we also think the reviewer raised a valid and good question. To empirically analyze the implication of using blank images, we conducted additional experiments with randomly selected non-blank images. Specifically, for each instance in our dataset, we repeated the probing experiments for the CS-L scores, but instead of a blank image, we used a randomly selected image from the Visual Genome dataset, and we repeated the experiments three times. For this rebuttal, we conducted these experiments only on  LLaVA and InstructBLIP, which showed the highest commonsense CS-L scores when given blank images. We will conduct full experiments with other models when revising the paper. The results are shown below. We can see that for LLaVA, using randomly selected images gives similar CS-L scores as using blank images, showing that LLaVA answers commonsense questions correctly about 30% of the time, regardless of whether a blank image or a random image is given. For InstructBLIP, using randomly selected images gives lower CS-L scores (around 35%) than using blank images (around 40%), implying that a randomly selected image negatively affects InstructBLIP’s commonsense capability. However, we can also see that the CS-VL scores of InstructBLIP are anyway much lower (23.55%), indicating that when counter-intuitive images are given instead of random images, InstructBLIP still suffers a lot.
>
> |              | **Commonsense CS-L (blank-image)** | **Commonsense CS-L (random image, repeated 3 times)** | **Commonsense CS-VL (counter-intuitive image)** |
> |:------------:|:----------------------------------:|:--------------------------------------------------:|:-----------------------------------------------------:|
> |     LLaVa    |               30.06%               |                       30.95%                       |                        27.92%                         |
> |     LLaVa    |                                    |                       30.51%                       |                                                       |
> |     LLaVa    |                                    |                       31.58%                       |                                                       |
> | InstructBLIP |               39.96%               |                       34.92%                       |                        23.55%                         |
> | InstructBLIP |                                    |                       35.44%                       |                                                       |
> | InstructBLIP |                                    |                       34.73%                       |                                                       |
>
>
> ### Question B
>
> We thank the reviewer for this suggestion. In this work, we did not use common images because there has been other work evaluating the general object/attribute/relation recognition capabilities of VLMs [1]. Due to limited time for rebuttal, we only managed to create 50 common images (e.g., yellow bananas) using Bing Image Creator. We used LLaVA and InstructBLIP to perform some preliminary experiments. We found that when these models were presented with common images and asked about commonsense attributes/spatial relations (e.g., “in general, is the color of a banana normally yellow”), these two models performed better (i.e., giving consistent commonsense answers) compared with the performance when the models were presented with counter-intuitive images. This is not surprising because the common images given reinforce the commonsense knowledge. Meanwhile, when presented with common images and asked about the specific attributes/spatial relations shown in the specific images, the models often struggled to give consistent and correct answers. We believe more systematic investigation is needed, and we will include a thorough experiment and analysis on this when revising the paper.
>
> ### Question C
>
> Yes, we will make our code and dataset publicly available.
>
>
> [1] Li et al., “Evaluating Object Hallucination in Large Vision-Language Models.” https://arxiv.org/abs/2305.10355

---

### Meta-Review · Area_Chair_vkVG · 2023-09-18

**Recommendation:** 3

**Metareview:**

This paper introduce a partially machine-generated new dataset, named ROME, to evaluate the reasoning capability of VLMs to correctly interpret counter-intuitive content. All reviewers agreed that evaluating VLMs under counter-intuitive content is interesting. However, the reviewers also pointed out that the scale of the dataset is relatively small and raised some concerns on the annotation process.

---

### Decision · Program_Chairs · 2023-10-07

**Decision:**

Accept-Findings

**Comment:**

This paper introduce a partially machine-generated new dataset, named ROME, to evaluate the reasoning capability of VLMs to correctly interpret counter-intuitive content. All reviewers agreed that evaluating VLMs under counter-intuitive content is interesting. However, the reviewers also pointed out that the scale of the dataset is relatively small and raised some concerns on the annotation process.